# Histone Deacetylase 3 Inhibitor Alleviates Cerebellar Defects in Perinatal Hypothyroid Mice by Stimulating Histone Acetylation and Transcription at Thyroid Hormone-Responsive Gene Loci

**DOI:** 10.3390/ijms23147869

**Published:** 2022-07-17

**Authors:** Alvin Susetyo, Sumiyasu Ishii, Yuki Fujiwara, Izuki Amano, Noriyuki Koibuchi

**Affiliations:** Department of Integrative Physiology, Gunma University Graduate School of Medicine, Maebashi 371-8511, Japan; m1820601@gunma-u.ac.jp (A.S.); y-fujiwara@gunma-u.ac.jp (Y.F.); iamano-lj@gunma-u.ac.jp (I.A.); nkoibuch@gunma-u.ac.jp (N.K.)

**Keywords:** histone deacetylase 3, inhibitor, hypothyroidism, cerebellum, organogenesis

## Abstract

Perinatal hypothyroidism impairs cerebellar organogenesis and results in motor coordination defects. The thyroid hormone receptor binds to corepressor complexes containing histone deacetylase (HDAC) 3 in the absence of ligands and acts as a transcriptional repressor. Although histone acetylation status is strongly correlated with transcriptional regulation, its role in cerebellar development remains largely unknown. We aimed to study whether the cerebellar developmental defects induced by perinatal hypothyroidism can be rescued by treatment with a specific HDAC3 inhibitor, RGFP966. Motor coordination was analyzed using three behavioral tests. The cerebella were subjected to RT-qPCR and chromatin immunoprecipitation assays for acetylated histone H3. The treatment with RGFP966 partially reversed the cerebellar morphological defects in perinatal hypothyroid mice. These findings were associated with the alleviation of motor coordination defects in these mice. In addition, the RGFP966 administration increased the mRNA levels of cerebellar thyroid hormone-responsive genes. These increases were accompanied by augmented histone acetylation status at these gene loci. These findings indicate that HDAC3 plays an important role in the cerebellar developmental defects induced by perinatal hypothyroidism. The HDAC3 inhibitor might serve as a novel therapeutic agent for hypothyroidism-induced cerebellar defects by acetylating histone tails and stimulating transcription at thyroid hormone-responsive gene loci.

## 1. Introduction

Transcriptional regulation plays crucial roles in the development of mammalian embryos and newborns. The formation of complicated cerebellar structures and neuronal networks is regulated by the spatiotemporally specific expression of the developmental genes [1,2,3]. It is well established that the cerebellum is responsible for motor coordination regulation and learning [4,5]. Developmental defects in the cerebellum are usually associated with ataxia in humans and mice [6,7]. It currently appears that the cerebellum also plays important roles in cognition and emotion [4,5,8,9]. Considering the importance of the cerebellum and poor performance status in patients with cerebellar defects, it is worth trying to ameliorate cerebellar developmental defects by modulating transcriptional regulation.

Thyroid hormones play a pivotal role in cerebellar organogenesis in both mice and humans [1,10,11,12]. Perinatal hypothyroidism results in multiple cerebellar defects, including reduced cerebellar size, the disturbed dendritic arborization of Purkinje cells, the delayed proliferation and migration of granule cells, and retarded synaptogenesis. Thyroid hormone receptors (TRs) are localized in the nuclei. The TRs bind to the thyroid hormone-response element (TRE) on their target genes and regulate transcription in a ligand-dependent manner [13,14]. The TRs suppress the transcription of the target genes in the absence of ligands, and transcription is induced when thyroid hormones are present. Indeed, TRs are expressed in most types of cells in the cerebellum [15,16].

Histone acetylation is strongly correlated with transcriptional regulation by numerous transcription factors, including TRs [17,18]. The TRs are associated with nuclear receptor corepressor complexes that contain histone deacetylase (HDAC) 3 in the absence of ligands [19,20]. The deacetylation of histone tails by HDAC3 is responsible for transcriptional repression by unliganded TRs. Nuclear receptor corepressor complexes specifically contain HDAC3 and are not associated with other HDACs [21]. When thyroid hormones bind TRs, the corepressor complexes with HDAC3 dissociate, and coactivator complexes that contain histone acetyltransferase activity are recruited to TRs, stimulating histone acetylation and transcription [22,23]. Therefore, it is highly likely that the enzymatic activity of HDAC3 is responsible for the cerebellar developmental defects in hypothyroidism. However, the role of histone acetylation status in cerebellar development has not been fully studied.

In this study, we attempted to rescue cerebellar developmental defects in perinatal hypothyroid mice by abrogating the enzymatic activity of HDAC3 using a specific inhibitor, RGFP966. The effects were evaluated by morphological and behavioral studies. Additionally, the mRNA levels and histone acetylation statuses were analyzed in vivo for TR-target genes.

## 2. Results

### 2.1. HDAC3-Inhibitor Treatment Increased Body Weight and Cerebellar Weight in Perinatal Hypothyroid Mice

We first tried to confirm that the specific HDAC3 inhibitor, RGFP966, antagonizes TR-dependent repression, although HDAC3 was reported to be responsible for transcriptional repression by unliganded TR. The addition of the RGFP966 relieved transcriptional repression by unliganded TR in a luciferase-based transcriptional reporter assay compared to vehicle treatment (Figure 1a). These results led us to study the in vivo effects of this inhibitor on hypothyroid subjects. We chose the mouse cerebellar developmental defect induced by perinatal hypothyroidism as a model because it causes symptoms that are well studied. The cerebella are mainly organized after birth in rodents. We induced hypothyroidism and consequent cerebellar developmental defects in the perinatal male mice by propylthiouracil (PTU) treatment and tried to rescue the phenotypes by the postnatal administration of RGFP966 (Figure 1b). For comparison, a group of hypothyroid littermates was treated with thyroxine (T_4_) to mimic euthyroid controls.

Consistent with previous reports, perinatal hypothyroidism resulted in reduced body weight. Interestingly, the RGFP966-treated hypothyroid mice gained more weight than the vehicle-treated hypothyroid mice (Figure 1c). Next, we measured the cerebellar weight on postnatal days (PND) 7 (Figure 1d–f) and 14 (Figure 1g–i). The cerebellar weights were lower in the vehicle group compared to the T_4_ group. The treatment with RGFP966 increased the weights (Figure 1d,g). The weights of the brains without cerebellum were similar among the three groups (Figure 1e,h). Therefore, the ratio of cerebellar weight to whole-brain weight was significantly increased in the inhibitor group compared to the vehicle group (Figure 1f,i).

### 2.2. HDAC3 Inhibitor Alleviated Cerebellar Morphological Defects in Perinatal Hypothyroid Mice

Next, we studied whether the increased cerebellar weight in the RGFP966-treated hypothyroid mice was associated with morphological improvement. The cerebella of the perinatal hypothyroid mice were smaller than those of the T_4_-supplemented mice, which is consistent with previous reports [10] (Figure 2a,c,d,f). The fissures between the lobules were shallow and the lobular structures were immature in these mice. The treatment with RGFP966 improved both the sizes and the morphological appearances, indicating the mitigation of cerebellar defects (Figure 1b,e).

The delayed migration of granule cells from the external granule cell layer (EGL) to the internal granule cell layer was observed in the immature developing cerebella, including those due to hypothyroidism. Indeed, the EGL was thicker in the vehicle-treated hypothyroid group than in the T_4_-supplemented group on PND 7 (Figure 2g,i), indicating the delayed migration. The EGL was thinner in the RGFP966 group than in the vehicle group (Figure 2h), demonstrating the acceleration of the migration. The EGL had almost disappeared in all three groups by PND 14 (Figure 2j–l).

### 2.3. HDAC3 Inhibitor Alleviated Motor Coordination Defects in Perinatal Hypothyroid Mice

Motor coordination is an important and highly established function regulated by the cerebellum. Perinatal hypothyroidism results in motor coordination defects because of impaired cerebellar development. We evaluated the motor coordination in the mice using three behavioral analyses. First, the surface righting test showed that the RGFP966-group mice returned to the proper position faster than the vehicle-group mice (Figure 3a). Next, the negative geotaxis test demonstrated that the RGFP966-group mice turned to face up the slope faster than the vehicle-group mice (Figure 3b). Finally, the mice treated with the RGFP966 were able to stay on the rotarod for longer than the vehicle group mice (Figure 3c). Taken together, these results suggest that treatment with RGFP966 alleviates motor coordination defects in perinatal hypothyroid mice. On the other hand, the RGFP966 administration did not affect the results of the surface righting test in the euthyroid mice (Figure 3d). Furthermore, the administration of RGFP966 in the euthyroid mice induced slight, but significant, motor coordination disturbances, as assessed by the negative geotaxis test and the rotarod test (Figure 3e,f). These results indicate the distinct effects of the HDAC3 inhibitor on motor coordination in hypothyroid mice and in euthyroid mice.

### 2.4. HDAC3 Inhibitor Alleviated Reduction in mRNA Levels of Cerebellar Thyroid Hormone Receptor-Target Genes in Perinatal Hypothyroid Mice

HDAC3 suppresses the transcription of TR-target genes by binding to unliganded TR as a component of the nuclear receptor corepressor complex. Therefore, we studied the mRNA levels of the TR-target genes as well as those of the other genes in the cerebella of the RGFP966-treated mice on PND 7 (Figure 4a–g) and PND 14 (Figure 4h–n).

On PND 7, the changes in mRNA levels were not statistically significant for many of the genes tested. The *Pcp2* and *Hr* are well established TR-target genes in the cerebellum. Typical TREs were reported in these gene loci [24,25]. The results suggested that these genes might be upregulated by the administration of T_4_ or RGFP966, but the differences were not statistically significant (Figure 4a,b). The expression of *Ntf3* and *Rora* is known to be stimulated by thyroid hormones in the cerebellum, although TREs have not yet been identified in these gene loci. The mRNA levels of these genes were lower in the vehicle-treated hypothyroid mice, as expected. The *Ntf3* mRNA levels were significantly upregulated by the administration of RGFP966 (Figure 4c). There seemed to be a slight upregulation of *Rora* mRNA levels in the inhibitor group, but the differences were not statistically significant (Figure 4d). The expression levels of *Bdnf* and *Grm1* are associated with cerebellar development, but the involvement of thyroid hormones in the regulation of these genes is not established. A significant difference was not observed between the three groups for these genes (Figure 4e,f). The levels of *Gapdh* mRNA were similar among the three groups when the results were normalized by the amount of *Rn18s*, one of the most reliable normalizers in RT-qPCR [26] (Figure 4g).

The mRNA levels of the four genes known to be stimulated by thyroid hormone (*Pcp2*, *Hr*, *Ntf3,* and *Rora*) were significantly increased on PND 14, regardless of the presence or absence of TRE. The treatment with RGFP966 increased the mRNA levels of these genes (Figure 4h–k). On PND 14, the increases were not as evident as those on PND 7 for *Ntf3*. This was probably because *Ntf3* is induced at earlier stages of cerebellar organogenesis than other genes. The inhibitor treatment tended to increase the mRNA levels of *Bdnf* and *Grm1*, but the differences were not statistically significant (Figure 4l,m). The levels of *Gapdh* mRNA did not change through the administration of either RGFP966 or T4, which was similar to the results on PND 7 (Figure 4n).

### 2.5. HDAC3-Inhibitor Treatment Increased Histone Acetylation Levels at Cerebellar Thyroid Hormone Receptor-Target Gene Loci in Perinatal Hypothyroid Mice

Finally, we investigated whether the increase in mRNA levels of TR-target genes by RGFP966 treatment was associated with the facilitated acetylation status of the histone tails. The mouse cerebella on PND 14 were subjected to an in vivo chromatin immunoprecipitation (ChIP) assay for acetylated histone H3. The RGFP966 treatment significantly increased the levels of histone acetylation in the regions containing TREs in the promoter of the *Pcp2* and *Hr* genes in the hypothyroid mouse cerebella (Figure 5a,b). Interestingly, the histones showed a tendency of facilitated acetylation through supplementation with T_4_ at the *Pcp2* locus (*p* = 0.06 vs. vehicle group), but not at the *Hr* locus. The histone acetylation levels were not affected by the RGFP966 administration or T_4_ at the transcription start site of the *Gapdh* gene, whose mRNA levels were not changed by these treatments (Figure 4g,n and Figure 5c). These results suggest the specificity of RGFP966’s action for the target genes, including thyroid hormone-responsive genes.

## 3. Discussion

We found that multiple defects in the perinatal hypothyroid mice were alleviated by the postnatal administration of the HDAC3 inhibitor, RGFP966. The cerebella were larger and structurally more organized, in addition to facilitating postnatal growth in the RGFP966-treated mice, when compared to their vehicle-treated hypothyroid littermates. The brain sizes without the cerebella were notably similar among the vehicle, RGFP966, and T_4_ groups. This was probably because the cerebellum is mainly organized after birth in rodents, while other areas in the brain develop earlier [1,10]. These findings suggest that our procedure works effectively as a rescue experiment for hypothyroid subjects. In addition, we divided the littermates into three groups, and comparisons were made among these groups. We were therefore able to minimize the maternal effects or environmental influences that interfere with behavioral tests in developing rodents [27].

The cerebellum is involved in regulating multiple functions, of which motor coordination is one of the most important [4,5]. Developmental defects in the cerebellum are commonly associated with ataxia in humans and mice [6,7]. The rotarod test is a well established experiment to study motor coordination [28]. As their motor coordination is disturbed due to the impaired cerebellar development, perinatal hypothyroid mice are not able to stay on the rotarod as long as euthyroid mice [29]. However, it is difficult to apply this task to mice in the early postnatal period because they generally cannot stay on the rod. Many of the behavioral tests in neonatal mice depend on reflexes. The surface righting test and negative geotaxis test are used to assess motor coordination, although the results are affected by several factors [27,30]. Therefore, the combination of these three tests would allow us to evaluate motor coordination during the developmental period, although the results of the rotarod test are also affected by other factors, such as muscle strength and endurance [31]. Notably, the alleviation of the motor coordination defects in the hypothyroid mice by the treatment with RGFP966 was observed in all the tests. The effect of RGFP966 on other cerebellar functions, such as cognition and emotion [4,5,8,9], should be studied further.

HDAC3 is the main enzyme among HDACs that is involved in histone deacetylation and transcriptional repression by unliganded TR [19,20]. It would be a reasonable approach to inhibit the enzymatic activity of HDAC3 to rescue the impairments induced by hypothyroidism. Its inhibitor, RGFP966, alleviated hypothyroidism-induced cerebellar developmental defects both morphologically and functionally, and the improvement was associated with increased mRNA levels and histone acetylation levels in TR-target genes. Our data demonstrate the pivotal role of HDAC3 in cerebellar developmental defects by hypothyroidism in vivo. These findings also suggest that RGFP966 might serve as a novel therapeutic agent for cerebellar defects in hypothyroidism. Hormonal supplementation therapy is the primary choice in hypothyroid subjects, but it should be administered during the so-called “critical period” of brain development. Thyroid hormone replacement cannot reverse brain impairment after this period [1]. The hypothyroid mice were treated with RGFP966 while hormonal supplementation was effective in the present study. It would be interesting to determine whether RGFP966 is effective after the “critical period.” In addition, hypothyroidism during the embryonic stage also results in several neuropsychological defects, such as low intelligence quotient scores, in humans [32]. It would be worth testing the effects of RGFP966 on these defects in the future.

The in vivo functions of HDAC3 are often studied using conditional knockout mice [33] because conventional HDAC3-null mice are embryonically lethal [34,35]. However, the pharmacological inhibition of enzymatic activity is an additional and important approach to the study of the roles of HDAC3, because this deacetylase also exhibits some functions that are independent of HDAC activity [36,37]. It would be important to distinguish between the enzyme-dependent and activity-independent functions of HDAC3. The administration of an inhibitor would lead to the development of a novel therapeutic strategy. The high specificity of RGFP966 as an inhibitor of HDAC3 is well documented [38]. Compared to pan-HDAC inhibitors, which are already in clinical use for the treatment of malignancies [39,40], it is expected that treatment with RGFP966 would result in less-severe adverse effects due to its specificity.

There are some limitations to this study. First, only one dosage of RGFP966 was administered. It was almost impossible to add more groups to this study because comparisons needed to be made among the male littermates, although studying the dose-dependent effect of RGFP966 would have been important. It was very rare to obtain more than four males within one litter. Detailed studies of adverse effects would also have been important, since the euthyroid mice treated with the same dosage of RGFP966 showed slight, but significant, motor coordination disturbances. Furthermore, since TR is not the only transcription factor that mediates repression by HDAC3, some of the effects of RGFP966 could have been achieved through other transcription factors independently of TR. This speculation might suggest the treatment of cerebellar defects with other etiologies by RGFP966. The identification of these transcription factors is a subject for future work. In addition, a transcription-independent mechanism including the deacetylation of non-histone proteins [41] would also be involved. Finally, the degrees of behavioral abnormality, mRNA levels, and histone acetylation levels did not perfectly correlate with each other. Studies with multiple dosages of RGFP966 and other transcription factors might produce more findings, as mentioned above. In addition, other types of histone modification and DNA methylation might also be involved [42,43]. HDAC3 inhibition might also affect the non-genomic action of thyroid hormones [44,45]. However, it might be difficult to solve this query because significant factors are involved in cerebellar organogenesis, to varying degrees.

## 4. Materials and Methods

### 4.1. Luciferase-Based Transcriptional Reporter Assay

Cell culture, transfection, and luciferase reporter assays were performed as previously described [46], with some modifications. The CV-1 cells were grown in DMEM supplemented with 10% fetal bovine serum (FBS). Cells were divided into 96-well plates at sub-confluency 24 h before transfection. Cells were transfected with a firefly luciferase reporter plasmid carrying direct-repeat type TRE (DR4-Luc), expression vectors for TRα and retinoid X receptor α, and CMV-β-galactosidase plasmid using TransIT-X2 dynamic delivery system (Mirus Bio, Madison, WI, USA), according to manufacturer’s instructions. The medium was changed to DMEM with 10% charcoal-stripped FBS the next day, and cells were further incubated overnight with either 10 nM of triiodothyronine (T_3_, Merck, Darmstadt, Germany), 1 μM of RGFP966 (Cayman Chemical, Ann Arbor, MI, USA), or vehicle (DMSO). Cell monolayers were lysed with 25 mM potassium phosphate (pH 7.8) containing 15 mM MgSO_4_, 2 mM EGTA, 1% Triton X-100, and 1 mM dithiothreitol. The luciferase activity was measured in 25 mM glycylglycine (pH 7.8) containing 15 mM MgSO_4_, 2 mM EGTA, 25 mM potassium phosphate, 1 mM dithiothreitol, 1 mM ATP, and 0.2 mM D-luciferin. Using 25 mM potassium phosphate, 1 mM MgCl_2_, and 1× Galacton-Plus Substrate (Thermo Fisher Scientific, Waltham, MA, USA), β-galactosidase activity was measured. The luciferase activity was normalized by β-galactosidase activity.

### 4.2. Animals

The animal study protocol was approved by the Institutional Review Board of Gunma University (protocol code 19-073). Pregnant C57BL/6J mice and pups were treated with 250 ppm of PTU (Merck, Darmstadt, Germany) in drinking water from E 14 to PND 21. Only litters with more than three males were used. Male-littermate pups were divided into three groups to minimize the maternal or environmental effects and were given daily subcutaneous injection from PND 3 up to PND 21. One group was given 10 mg/kg body weight (BW) of RGFP966. The vehicle control group was injected with the corresponding amount of DMSO, since RGFP966 was dissolved in DMSO. The last group was administered 20 μg/kg BW of T_4_ (Merck, Darmstadt, Germany), which is the amount used to rescue PTU-induced hypothyroidism in perinatal mice [47]. The T_4_ group was also injected with DMSO because the effect of DMSO should not be ignored during organogenesis. All pups were weighed every day to determine the amount of injections. Mice were heavily anesthetized with an intraperitoneal injection of 5 mL/kg BW of ketamine/xylazine mixture (22.5 mg/mL ketamine and 1 mg/mL xylazine in saline) and sacrificed by decapitation before collecting tissue samples.

### 4.3. Histological Studies

On PND 7 and PND 14, mice were fixed by perfusion with 4% paraformaldehyde, and cerebella were subjected to cryosectioning. Sagittal sections were stained with hematoxylin and eosin. Images were captured using a BZ-9000 microscope (Keyence, Osaka, Japan).

### 4.4. Behavioral Studies

According to previous reports [27,30], the surface righting reflex was tested from PND 3 to PND 12, and the negative geotaxis test was performed on PND 7. Pups were gently placed in the supine position on a flat surface, and the time to flip onto their bellies was measured for surface righting. A negative geotaxis test was performed by placing the mice on a 45-degree slope with their heads facing downwards. The time until the mice turn to face up the slope was recorded. Both tests were repeated three times and the average times were analyzed.

On PND 21, the mice were placed on the LE8500 rotarod unit (Panlab, Barcelona, Spain) and the time taken to fall from the rod was measured [28,29]. An acceleration program from 4 to 40 rounds per minute (rpm) for 2 min followed by a constant 40 rpm rotation mode for 1 min was used after adaptation at a constant speed of 4 rpm for 1 min. Five trials were performed, and the average time was used for analyses.

### 4.5. Quantitative RT-PCR

Total RNA was extracted from mouse cerebella on PND 7 and PND 14 by QIAzol (Qiagen, Hilden, Germany), according to the manufacturer’s instructions. Total RNA (0.5 μg) was used for cDNA synthesis by ReverTra Ace qPCR RT Master Mix (TOYOBO, Osaka, Japan). Quantitative PCR was performed using the THUNDERBIRD SYBR qPCR Mix (TOYOBO, Osaka, Japan), according to the manufacturer’s instructions. As one of the most reliable normalizers in RT-qPCR [26], the mRNA levels of each gene were normalized to those of *Rn18s*, and were expressed as the relative amount compared to the control group. The primers used for quantitative RT-PCR are shown in Table 1.

### 4.6. In Vivo ChIP Assay

On PND 14, mouse cerebella were snap-frozen in liquid nitrogen and crushed in frozen mortars on liquid nitrogen. The crushed samples were subjected to cross-linking in 1% paraformaldehyde for 10 min at room temperature and subsequent ChIP assay using ChIP Assay Kit (Merck, Darmstadt, Germany), according to the manufacturer’s instructions. One cerebellum was used for each immunoprecipitation. Anti-acetyl-Histone H3 antibody (5 μg) (Merck, Darmstadt, Germany) or control IgG were used for immunoprecipitation. The DNA samples were analyzed by quantitative PCR using the THUNDERBIRD SYBR qPCR mix (TOYOBO, Osaka, Japan), according to the manufacturer’s instructions. The results were normalized by the amount of input. The amounts of immunoprecipitated DNA by acetyl-Histone H3 antibody were divided by those of DNA bound to control IgG and were expressed as the relative amount compared to the control group. The primers used for quantitative PCR are shown in Table 2.

### 4.7. Statistical Analyses

One-way analysis of variance (ANOVA) or two-way ANOVA was followed by a post hoc Tukey’s HSD test. The student’s *t*-test was used for comparison between two groups. The analyses were performed using JMP (SAS Institute, Cary, NC, USA).

## 5. Conclusions

In summary, the pharmacological inhibition of HDAC3 enzymatic activity by RGFP966 alleviated the cerebellar morphological defects in perinatal hypothyroid mice. Functionally, the mice treated with RGFP966 exhibited better motor coordination than the hypothyroid control mice. The inhibitor treatment increased the mRNA levels of the TR-target genes in the hypothyroid cerebella. In addition, the increases in mRNA levels were associated with facilitated histone acetylation status in the loci of the TR-target genes (Figure 6). These results demonstrate the important roles of HDAC3 in the cerebellar developmental defects induced by perinatal hypothyroidism. The results of the present study suggest that the HDAC3 inhibitor might serve as a novel therapeutic strategy for cerebellar developmental defects.

## Figures and Tables

**Figure 1 ijms-23-07869-f001:**
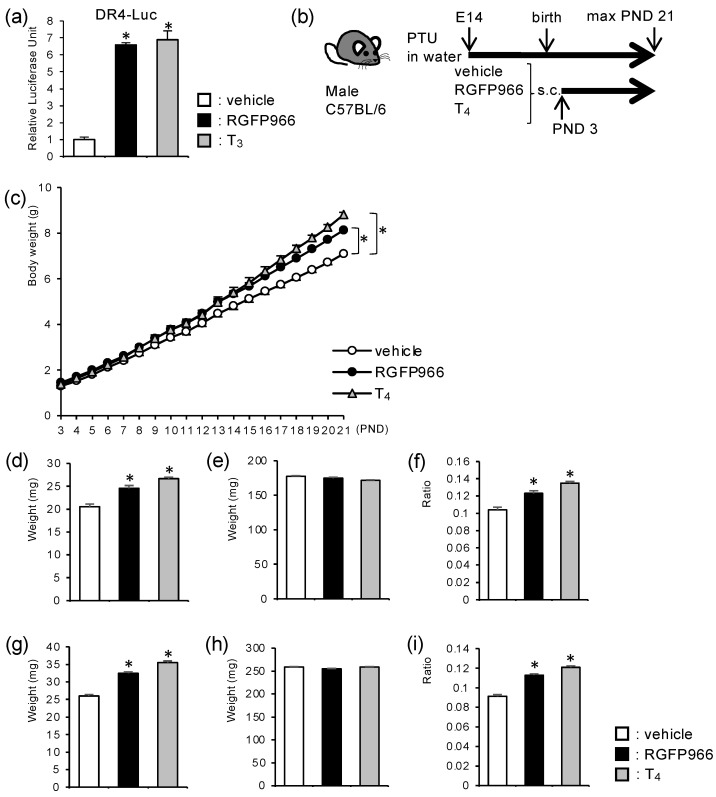
HDAC3-inhibitor treatment increased body and cerebellar weights in perinatal hypothyroid mice. (**a**) HDAC3 inhibitor, RGFP966, relieved transcriptional repression by unliganded thyroid hormone receptors in a luciferase-based transcriptional reporter assay. (**b**) Schematic representation of the experimental schedule. C57BL/6 mice were treated with PTU in drinking water starting from embryonic day (E) 14. Male littermates were divided into three groups and were subjected to daily subcutaneous injection (s.c.) of either vehicle, RGFP966, or T_4_ from PND 3. (**c**) RGFP966 treatment increased body weight in perinatal hypothyroid mice. (**d**–**i**) RGFP966 treatment increased cerebellar weight in perinatal hypothyroid mice on (**d**–**f**) PND 7 and (**g**–**i**) PND 14. (**d**,**g**) Cerebellar weight, (**e**,**h**) the weight of the brain without a cerebellum, and (**f**,**i**) cerebellum/whole brain ratio are shown. Data are expressed as the mean ± SEM. *: *p* < 0.01 vs. vehicle group by Tukey’s HSD test. *n* = 13 for each group.

**Figure 2 ijms-23-07869-f002:**
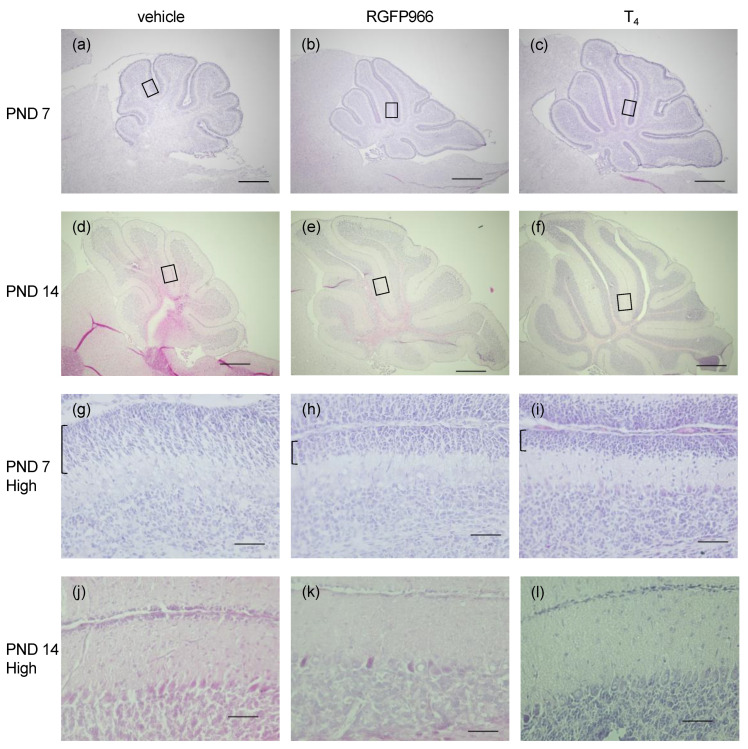
HDAC3 inhibitor alleviated cerebellar morphological defects in perinatal hypothyroid mice. Cerebellar sagittal sections were subjected to hematoxylin and eosin staining. Representative images from (**a**–**c**) PND 7 and (**d**–**f**) PND 14 using a 4× objective are shown. Scale bars: 500 μm. The rectangular areas in lobule V shown in (**a**–**f**) were further analyzed by a 40× objective (**g**–**l**), respectively. External granule cell layers at PND 7 are indicated by square brackets (**g**–**i**). Scale bars: 50 μm. Images from (**a**,**d**,**g**,**j**) vehicle group, (**b**,**e**,**h**,**k**) HDAC3 inhibitor RGFP966 group, and (**c**,**f**,**i**,**l**) T_4_ group are shown.

**Figure 3 ijms-23-07869-f003:**
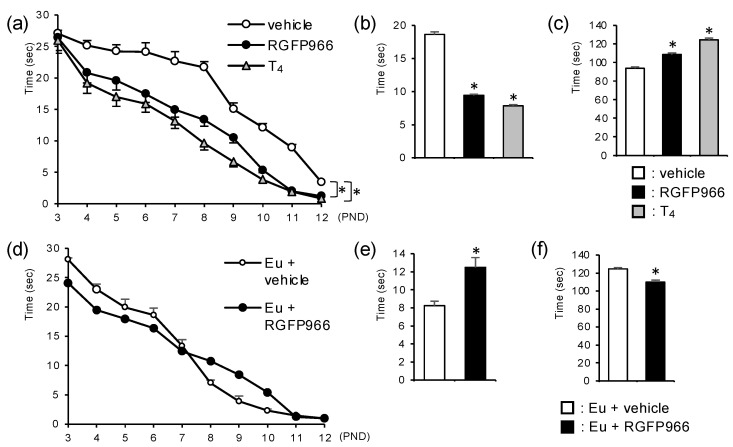
HDAC3 inhibitor alleviated motor coordination defects in perinatal hypothyroid mice (**a**–**c**). Motor coordination in PTU-treated mice was assessed by (**a**) surface righting test, (**b**) negative geotaxis test on PND 7, and (**c**) rotarod test on PND 21 for vehicle group, HDAC3 inhibitor RGFP966 group, and T_4_ group. Data are expressed as the mean ± SEM. *: *p* < 0.01 vs. vehicle group by Tukey’s HSD test. *n* = 13 for each group. (**d**–**f**) Male euthyroid (Eu) C57BL/6 mice were similarly treated with either vehicle or RGFP966 and were subjected to (**d**) surface righting test, (**e**) negative geotaxis test on PND 7, and (**f**) rotarod test on PND 21. Data are expressed as the mean ± SEM. *: *p* < 0.01 vs. vehicle group by Student’s *t*-test. *n* = 10 for each group.

**Figure 4 ijms-23-07869-f004:**
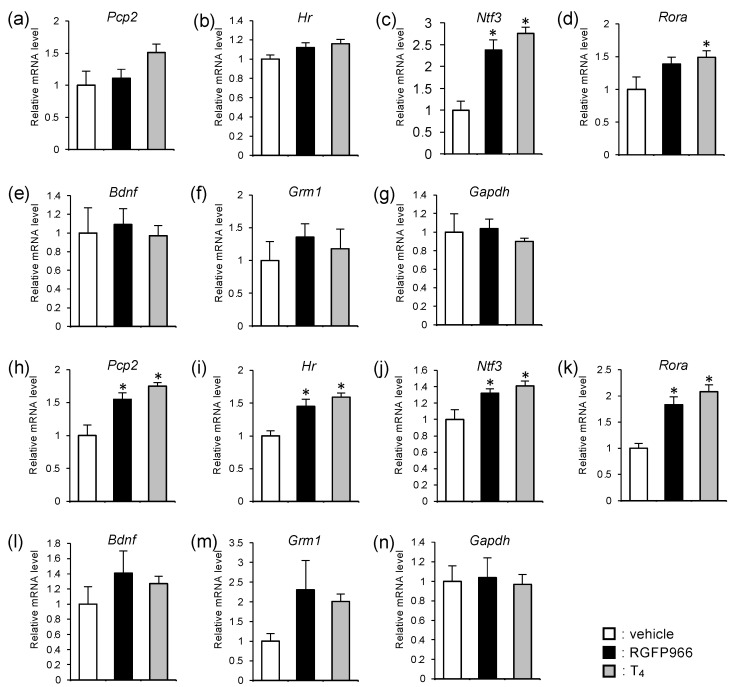
HDAC3 inhibitor alleviated reduction in mRNA levels of TR-target genes in perinatal hypothyroid mouse cerebella. Relative mRNA levels for each gene in the cerebellum on (**a**–**g**) PND 7 and (**h**–**n**) PND 14 were analyzed by RT-qPCR. The results were normalized by the amount of *Rn18s* and were expressed as the relative amount compared to the control group. Data are expressed as the mean ± SEM. *: *p* < 0.05 vs. vehicle group by Tukey’s HSD test. *n* = 8 for each group.

**Figure 5 ijms-23-07869-f005:**
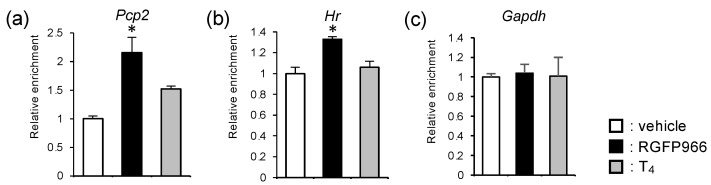
HDAC3-inhibitor treatment increased histone acetylation levels at TR-target gene loci in perinatal hypothyroid mouse cerebella. (**a**–**c**) Histone acetylation status was analyzed for each gene locus by in vivo ChIP assay for acetylated histone H3 using PND 14 cerebella. The results were expressed as the relative enrichment of acetylated histone H3 compared to the control group for each gene. Data are expressed as the mean ± SEM. *: *p* < 0.05 vs. vehicle group by Tukey’s HSD test. *n* = 3 for each group.

**Figure 6 ijms-23-07869-f006:**
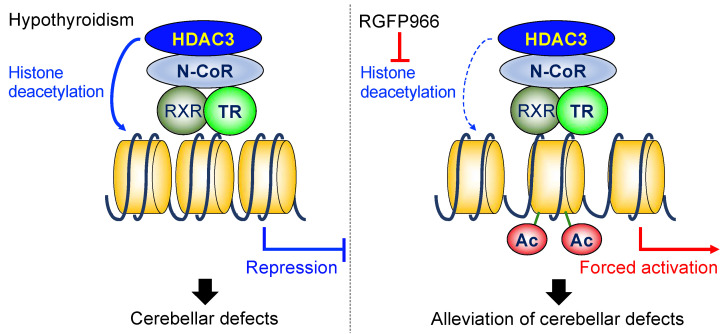
Diagrammatic representation of the working model. N-CoR: nuclear receptor corepressor, RXR: retinoid X receptor, Ac: acetylation of the histone tails.

**Table 1 ijms-23-07869-t001:** Primers for RT-qPCR.

Gene Symbols	Forward	Reverse
*Pcp2*	GGATGGAATGCAGAAACGAC	GTTCCTGCGGAAGCTGAGT
*Hr*	TTGGCCCTTGTAGGAAATGC	TTTCAGCTTGGTGTGATGGC
*Ntf3*	TGCCACGATCTTACAGGTGA	AGTCTTCCGGCAAACTCCTT
*Rora*	TCCTTCACCAACGGAGAGAC	CCAGGTGGGATTTGGATATG
*Bdnf*	ATCCAAAGGCCAACTGAAGC	ATTGGGTAGTTCGGCATTGC
*Grm1*	CGAAGGCTATGAGGTGGAAG	AAGGATTCCTCGTGTTGGTG
*Gapdh*	TGCGACTTCCAACAGCAACTC	ATGTAGGCCATGAGGTCCAC
*Rn18s*	CGGCTACCACATCCAAGGAA	GGGCCTCGAAAGAGTCCTGT

**Table 2 ijms-23-07869-t002:** Primers for chromatin immunoprecipitation.

Targets	Forward	Reverse
*Pcp2* (TRE)	CTCCCCCTACACCCTAGCCTTTTA	CTAGAAGGTCTGAGCCTCCCTCTC
*Hr* (TRE)	TCCTGAGAGCTCTGGTCTAGC	CCTGACCTCTGGCTCCTG
*Gapdh* (TSS)	CTAGGACTGGATAAGCAGGGCGG	TGGAACAGGGAGGAGCAGAGAG

TSS: transcription start site.

## Data Availability

The data presented in this study are available on request from the corresponding author.

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
