# Peer review of "Histone Deacetylase 3 Inhibitor Alleviates Cerebellar Defects in Perinatal Hypothyroid Mice by Stimulating Histone Acetylation and Transcription at Thyroid Hormone-Responsive Gene Loci"

_ijms, 2022, doi:10.3390/ijms23147869_

Round 1

Reviewer 1 Report

The paper “Histone deacetylase 3 inhibitor alleviates cerebellar defects in 2 perinatal hypothyroid mice by stimulating histone acetylation 3 and transcription at thyroid hormone responsive gene loci” aims to show interesting results on the activity of HDAC3 inhibitor RGFP966 in mice in realtion to cerebeller development. This manuscript is well written in all its parts, material and methods are properly detailed and the results are clearly shown and discussed. As far as I'm concerned it therefore deserves to be published in IJMS as it is

Reviewer 2 Report

Susetyo et al. investigated the effects of histone deacetylase 3 (HDAC3) inhibition on perinatal hypothyroidism-induced cerebellar defects. Upon treatment of mouse models with the selective HDAC3 inhibitor RGFP966, they found that RGFP966 reduced hypothyroidism-induced defects including changes in body weight and cerebellar weight, as well as morphological and functional defects. They also performed molecular analyses and found RGFP966 reduced mRNA levels of cerebellar thyroid hormone receptor-target genes and increased histone acetylation at these loci. The manuscript is well-written and would benefit from this minor addition:

-      - The authors may need to add a diagrammatic representation to serve as a working model at the end of the paper. This figure should summarize the role of HDAC3 in hypothyroidism-induced cerebellar defects, and the effects observed with RGFP966 in the study. This working model will help guide readers through the main findings of the study. 
